# Characterization of the Water Shortage Effects on Potato Tuber Tissues during Growth Using MRI Relaxometry and Biochemical Parameters

**DOI:** 10.3390/plants11151918

**Published:** 2022-07-25

**Authors:** Ghina Hajjar, Stéphane Quellec, Sylvain Challois, Lydia Bousset-Vaslin, Gisèle Joly, Christophe Langrume, Carole Deleu, Laurent Leport, Maja Musse

**Affiliations:** 1UR OPAALE, INRAE, 17 Avenue de Cucillé, CS 64427, 35044 Rennes, France; ghina.hajjar@inrae.fr (G.H.); stephane.quellec@inrae.fr (S.Q.); sylvain.challois@inrae.fr (S.C.); 2UMR IGEPP, INRAE, Institut Agro-Agrocampus Ouest, Université de Rennes 1, Domaine de la Motte, 35653 Le Rheu, France; lydia.bousset@inrae.fr (L.B.-V.); christophe.langrume@inrae.fr (C.L.); carole.deleu@univ-rennes1.fr (C.D.); 3Germicopa, 1 allée Loeiz Herrieu, 29334 Quimper, France; gisele.joly@germicopa.fr

**Keywords:** magnetic resonance imaging, *Solanum tuberosum*, multi-exponential transverse relaxation, water stress

## Abstract

The potato is one of the most cultivated crops worldwide, providing an important source of food. The quality of potato tubers relates to their size and dry matter composition and to the absence of physiological defects. It depends on the spatial and temporal coordination of growth and metabolic processes in the major tuber tissues: the cortex, flesh and pith. In the present study, variations in the biochemical traits of each of these tissues were investigated during tuber growth under optimal and water-deficit conditions. MRI relaxometry was used as a non-invasive and quantitative method to access information on cellular water status. The presence of slight but significant variations in organic compound contents quantified in the cortex and flesh revealed a tissue-dependent metabolic pattern. The T_2_ and relative I_0_ of the bi-exponential relaxation signal allowed a distinction to be made between the pith and the cortex, whereas the flesh could be differentiated from these tissues only through its relative I_0_. T_2_ values did not vary significantly during tuber development, in accordance with the typical growth pattern of tubers, but were shown to be sensitive to water stress. The interpretation of the multi-exponential transverse relaxation times is discussed and could be further developed via microscopic analysis.

## 1. Introduction

The potato is one of the most important crops in a number of countries across the world, supplying many important nutrients to the human diet. Potato tubers contain around 80% water and around 10–25% of FW of starch depending on cultivar [1]. Tuber quality relates to size, dry matter composition and the absence of external and internal physiological defects. It is determined by the spatial and temporal coordination of growth and metabolic processes, in particular tuber tissues. Potato tubers are complex organs comprising three major tissues: the flesh (perimedullary zone), the cortex and the pith. Cell division and enlargement in these three tissues during tuber growth occur according to specific tissue-related kinetics [2]. The differentiation of the cortex and pith occurs when the cells in the apex divide, whereas that of the flesh tissue results from the division of the procambial cells [3]. It has been reported that the onset of tuber formation occurs when cells divide longitudinally to the stolon axis [2]. Tuber growth is then associated with the randomly oriented division and enlargement of cells in the flesh until the final size is reached [2]. Since cell size does not increase in the course of tuber growth, the tuber final size is not related to the size of the flesh cells but depends on the number of cells [4]. Round cells have been observed in the flesh and cortex, those located in the flesh tissue being 1.5 to 2.5 times the size of those in the cortex [5]. The pith tissue is made up of tube-shaped cells orientated along the stolon-top axis with cell transverse-section sizes between those found in the cortex and flesh. In addition to these morphological developments, the formation of tubers also involves the supply of photosynthates from the leaves followed by the biochemical processes that lead to the formation and storage of starch. It has been reported that, during tuber growth, sucrose is provided through phloem symplastic unloading rather than the apoplastic unloading observed before tuberization [6]. During tuber growth, starch accumulates in the flesh and cortex tissues, whereas relatively few starch grains are observed in the pith [7].

Due to its shallow and sparse root system [8], the potato plant is sensitive to water deficit. It responds to drought by initiating complex physiological and metabolic processes. Tuber development is impacted by drought, mainly during the tuber bulking and tuber ripening stages [9]. The relationship between yield components (i.e., the number and size of tubers) depends in a complex way on the timing, duration and severity of the water deficit. In some cases, the number and size of tubers may compensate each other [10]. In addition to tuber size, the water regime may affect the tuber weight and starch content [9] that reflect tuber quality. A differential effect of water regime on tuber tissues was to be expected as a result of their specific composition and growth rate. However, to the best of our knowledge, no such data are available. It is therefore necessary to characterize the individual tissues to gain a better understanding of the effects of drought on tuber quality.

Because it can provide quantitative information on water dynamics and structural features at multiple length scales (from nm to few centimeters) and in a non-destructive way, quantitative magnetic resonance imaging (MRI) offers a valuable approach to the characterization of plant organs. In the case of organs with high water content (fruits, tubers, leaves), the MRI signal is mainly defined by longitudinal (T_1_) and transverse (T_2_) relaxation parameters and the self-diffusion coefficient of water molecules. In a number of studies, MRI was used to visualize and quantify morphological features and internal defects in plant organs [11,12]. Additionally, MRI relaxometry has been used to access information on water status, subcellular compartmentalization of water, and membrane permeability/integrity in plant organs [13]. Indeed, in the main cell compartments of plant tissues, water is characterized by different T_2_ and T_1_ relaxation times that are governed by water mobility and the chemical exchange of water protons with macromolecules (polysaccharides and proteins) and solid surfaces. Since the diffusion exchange of water molecules between compartments separated by a membrane is relatively slow, the relaxation signal is generally multi-exponential, although the process of exchange tends to produce a partially-averaged signal from the water in these compartments. The relaxation signal thus reflects water compartmentalization and is impacted by membrane status and the size of the compartments [14]. The diffusion of water in the random magnetic field gradients that is generated by susceptibility-induced inhomogeneities derived from gas-filled intercellular spaces also affects relaxation times. Several NMR and MRI studies [14,15,16] have sought to interpret the relaxation data of specific plant tissues in terms of their structure and composition. Among these studies, only two concern tubers [17,18]. Further investigation is therefore required to establish interpretations of relaxation parameters that take into account the specificities of tuber tissues, in particular their high starch content.

In the present study, we investigated the variation in the biochemical traits of tissues during tuber growth under optimal and water-deficit conditions and explored their relationship to changes in transverse relaxation times. Analysis covered the major tuber tissues, i.e., the cortex, the flesh and the pith. The first aim was to characterize the effects of water deficit on the spatial and temporal progress of the biochemical processes occurring in tuber tissues during growth. The second aim was to improve interpretation of the changes in relaxation parameters in tuber tissues so that MRI parameters could be proposed as biomarkers for the assessment of tuber development.

## 2. Results

### 2.1. Changes in Physiological Parameters

Under the water deficit treatment, a loss of 230 g of tuber fresh weight per plant was recorded compared with plants under control conditions. This corresponded to a 63% decrease in final tuber yield measured at 87 days after shoot emergence (DASE). For the three largest tubers (the only tubers selected for further analysis of the effects of the treatment), water stress had an impact on tuber size during the period from 30 to 60 DASE (Figure 1). After 60 DASE, tuber weight did not change significantly, reaching a continuous plateau through to 87 DASE (data not shown).

The evolution of the biochemical parameters in the flesh and cortex is shown in Figure 2 and Figure 3 and Table 1, Table 2 and Table 3. Note that the pith was not subjected to biochemical analysis (Section 4). For the control tubers, the average pH value of the cortex was 6.4 at 32 DASE and differed significantly (Table 1) from that of the flesh, which had an average of 6.2 (Figure 2A). The pH was stable between 32 and 46 DASE and significantly decreased to 5.9 ± 0.1 at 60 DASE in both tissues (Table 2). Overall, severe water deficit (SWD) induced a slight but significant decrease in pH except in the flesh at 32 DASE (Table 3). However, water stress did not modify the pattern of pH evolution during tuber filling (i.e., stable until 45 DASE and then decreasing).

As shown in Figure 2B, the redox potential Eh (adjusted for pH = 7) in the cortex of control tubers was 284 ± 21 mV at 32 DASE with no significant difference compared with values measured in the flesh (272 ± 19 mV) (Table 1). In both tissues, Eh values significantly decreased to 224 ± 22 mV in the cortex and 223 ± 14 mV in the flesh at 46 DASE and then to 174 ± 14 and 169 ± 10 mV, respectively, at 60 DASE. Water stress did not have an impact on Eh values in tuber tissues (Table 3) and the same pattern of Eh evolution during filling was observed in the tissues of SWD tubers.

Water content (WC) was approximately 86% of tuber fresh weight (FW) when measurements began (32 DASE), falling to below 80% at 60 DASE in both the cortex and flesh of control tubers (Figure 3A). However, no significant difference was observed between the measurement dates, probably due to the heterogeneity of the samples at 60 DASE (Table 2). In both tissues, water stress induced a significant decrease in WC only at 46 DASE (Table 3).

As expected, starch was the main carbohydrate found in the potato tuber tissues. Its evolution during growth in the cortex and flesh of both control and SWD tubers is shown in Figure 3B. At 32 DASE, the starch content of the control tuber cortex (58 ± 4% of dry weight, DW) was not significantly different from that of the flesh. In both tissues, starch content significantly increased to approximately 70% of DW at 46 DASE and reached 76 ± 3 and 77 ± 3% of DW, respectively, at 60 DASE. Other carbohydrates were identified in the potato tuber tissues (sucrose, a disaccharide, fructose and glucose, two monosaccharides) as well as polyols (myo-inositol). In the cortex of control tubers, glucose and fructose contents were 133 ± 48 and 21 ± 11 µmol/g of DW, respectively, at 32 DASE and there was no significant difference in content levels in the flesh of these tubers (Figure 3C,D). Glucose and fructose contents both decreased during filling. At 60 DASE, the greatest decrease was observed in fructose, at 95%, whereas a decrease of 75% was observed for glucose. However, differences were observed between content levels in the flesh and the cortex. As shown in Figure 3E, sucrose content was 93 ± 4% µmol/g of DW in the cortex of control tubers at 32 DASE, significantly lower than its content in the flesh (133 ± 29 µmol/g of DW). Sucrose content also decreased during tuber filling but its decrease was more pronounced in the flesh (57%) than in the cortex (45%). Water deficit had no significant impact on the carbohydrates profile (starch, glucose, fructose and sucrose), with the same pattern of evolution during tuber filling being observed in both tissue types for the SWD tubers (Figure 3B–E).

The sucrose-to-hexose ratio, which is generally considered an indicator for water stress, is shown in Figure 3F. At 32 DASE, this ratio was equal to one in the cortex of the control tubers with no significant difference from the ratio measured in the flesh. This means that in both tissues of the control tubers sucrose and hexose (mainly glucose) contents were equivalent. A slight increase was observed in both the cortex and the flesh of the control tubers, with the sucrose-to-hexose ratio not exceeding two. For both tissue types, water deficit had no impact on the sucrose-to-hexose ratio at the beginning of the filling phase (32 DASE). However, for SWD tubers, an important increase was observed between 46 and 60 DASE, with ratio values above 6 at the end of the filling phase.

Small amounts of myo-inositol in the potato tuber cortex and flesh were quantified (Figure 3G). For both tissue types in the control tubers, myo-inositol content was negligible. Its content levels increased at the late filling stages reaching values of approximately 2 µmol/g DW. By contrast, equivalent amounts of myo-inositol were already quantified in SWD tubers for both tissue types at 32 DASE and remained stable during the filling period.

In addition to carbohydrates and polyols, certain organic acids were also measured in the potato tuber tissues, namely citrate, malate, and quinate (Figure 3H). At 32 DASE, the total content of organic acids in the control cortex was 67 ± 8 µmol/g of DW, differing significantly from levels observed in the flesh (52 ± 4 µmol/g of DW). Levels decreased significantly in both tissues at 46 DASE, down to 44 ± 9 and 35 ± 5 µmol/g of DW, respectively, in the cortex and flesh, and remained stable until 60 DASE. The same pattern was observed in both the cortex and flesh of the SWD tubers. However, organic acid content levels were significantly higher in the cortex of SWD tubers compared with those of the control tubers on the three measurement dates.

### 2.2. Evolution of MRI Parameters

An example of T_2_-weighted MRI image (TE = 130 ms) acquired with Multi-Spin Echo sequence used for tissue segmentation is shown in Figure 4A. The signal from the pith was higher, indicating longer T_2_ time in this tissue. In the flesh and cortex, the signal had a similar intensity, indicating close T_2_ values. However, these two tissues could apparently be distinguished, because of a narrow and bright ring-shaped structure that probably corresponded to vascular bundles at the interface between the cortex and the flesh. Note that the cortex and the pith appeared as homogeneous regions, in contrast to the flesh, characterized by a more heterogeneous signal. This was observed for all tubers as shown in Appendix A. The T_2_ decay curves for the three different tissue types (cortex, flesh and pith) of both the control and SWD tubers were optimally fitted using a bi-exponential model that reflected two water pools in each tissue: (1) a component with T_2_ values below 100 ms and (2) a component with T_2_ values above 200 ms (Figure 5 and Appendix A).

The relative signal intensity of component (2) is shown in Figure 5A since this corresponds to the major water pool in the tissues. At 32 DASE, I_0_ of component (2) was 81 ± 2% in the cortex of the control tubers. This value (Table 4, Table 5 and Table 6) was significantly higher than that found in the flesh (76 ± 2%) and significantly lower than that in the pith (88 ± 2%). At the end of tuber filling (60 DASE), I_0_ of component (2) significantly decreased to 73 ± 4% in the cortex of the control tubers, whereas the decreasing trend in the flesh of control tubers (Figure 5A) was not significant. Accordingly, in the control tubers, from 46 DASE, the relative signal distributions between components (1) and (2) in the cortex and the flesh were similar but these differed from that measured in the pith. Water deficit did not change the evolution pattern of I_0_ (Table 4, Table 5 and Table 6).

At 32 DASE, component (1) T_2_ was 64 ± 17 ms for the cortex of the control tubers. This value was not significantly different from those measured for the flesh (61 ± 11 ms) or the pith (52 ± 15 ms) (Figure 5B and Table 4). Component (1) T_2_ remained stable during tuber filling (Table 5) and was not affected by water deficit (Table 6). By contrast, component (2) T_2_ was 268 ± 4 ms for the cortex of the control tubers at 32 DASE (Figure 5C). There was no significant difference in the control tubers between component (2) T_2_ values for the cortex and the flesh (Table 4). However, the component (2) T_2_ value for the pith was 353 ± 15 ms which was significantly higher (~25%) than that measured for both the cortex and the flesh. An increase in component (2) T_2_ values over the filling period was observed only for the flesh. However, the component (2) T_2_ values measured for the same tissue were significantly lower in the SWD tubers than in the control tubers. Indeed, water deficit induced a decrease of around 20% in all tissues throughout the filling period.

### 2.3. Global Analysis of Physiological and MRI Parameters

By combining all parameters measured in the cortex and the flesh of tubers on the three different dates (32, 46 and 60 DASE), it was possible to carry out a principal component analysis (PCA) and to propose an exploratory global interpretation of the effects of tissue type, water condition and growth phase. Here, two principal components were significant, explaining 65% of observed variance (Figure 6). The first principal component (PC1) accounted for 46% of the variance and made it possible to distinguish between observations by growth phase (Figure 6A). The most significant variables in this component were the content levels of myo-inositol, starch and other carbohydrates as well as redox potential (Figure 6C). Starch content correlated negatively with glucose, fructose and sucrose along the PC1 axis. Starch content also correlated negatively with component (2) I_0_, pH, redox potential and WC. Further, the second principal component (PC2) accounted for 19% of observed variance and made it possible to distinguish between observations by water regime (Figure 6B). The most significant variables in this component were the T_2_ values obtained from MRI measurements (Figure 6C). PCA analysis also showed that the difference between control and SWD conditions increased during tuber filling (Figure 6A,B). However, PCA did not make it possible to distinguish between the cortex and flesh of the potato tubers (Appendix A). The content levels for organic acids were almost the only discriminating factor between these two tissues (Figure 3H and Table 1, Table 2 and Table 3).

## 3. Discussion

### 3.1. Spatial Changes in Composition during Tuber Filling under Well-Irrigated and Stress Conditions

Tuber growth is sustained by the transport, unloading and utilization of photoassimilates as a substrate for cell respiration, division and enlargement and starch deposition [19]. Under optimal irrigation conditions, adequate leaf-water relations allow unrestricted CO_2_ fixation, resulting in the production of large tubers [20]. The results of the present paper confirm those previously obtained [20], showing the continuous growth of individual tubers from two weeks after stolon swelling and tuber initiation through top-kill. During this phase of linear tuber growth, tubers can differ considerably in growth rate, depending on their competing strength [21]. In the present study, the three tubers selected from each plant for analysis were the heaviest and of similar strength; consequently, no such variations were observed.

During tuber growth, the slight decrease in water content observed in the cortex and the flesh (Figure 3A) was mainly due to the increase in dry content associated with starch synthesis. Indeed, in potatoes, cell division and expansion is rapidly followed by massive starch deposition [22]. Thus, at the beginning of tuber growth, starch already comprised more than half of the total dry-matter component (the latter being around 8% of tuber fresh weight, Figure 3B), in accordance with the literature [22]. The fact that the starch content reached its maximum at this early stage of tuber development may be associated with the particular processes involved in tuber growth, where cell division and enlargement progress from the inner flesh towards the periphery are followed immediately by starch deposition [2,6]. This is consistent with the PCA analysis showing that starch content was one of the main parameters affecting the first PCA component (Figure 6C), explained by the developmental stage (Figure 6A). The present study is original in demonstrating that starch accumulation followed similar patterns in the cortex and flesh tissues. Although the starch content of the pith was not quantified due to experimental constraints, micrographs (Appendix A) indicated the starch content of the pith to be lower than that of the other potato tissues in line with Oey et al. [7].

Quantification of organic chemicals in the flesh tissue confirmed earlier results obtained for the whole tuber [6]. For instance, the low fructose content (Figure 3D) observed in the flesh throughout the period of tuber growth is associated with reported high sucrose synthase (SuSy) activity during tuberization [6]. The similar levels of the glucose and sucrose can be linked to the high sucrose hydrolytic activity that has been associated with tuber sink strength [4,6]. It can be noted that the decrease in the quantity of all soluble sugars (i.e., sucrose, glucose and fructose) from 45 DASE is probably explained by a dry-matter dilution effect linked to starch accumulation during this period. Differences between the cortex and flesh tissues in terms of their fructose and sucrose contents can be explained by differences in the carbon metabolism of these tissues. Contrary to expectation, the higher content of organic acids observed in the cortex when compared with the flesh (Figure 3H), produces a slightly higher pH (Figure 2A) rather than greater acidity.

The impact of water stress on tuber size (Figure 1) may be explained by the major change in source-sink relations brought about by, on one hand, a marked decrease in source activity with the stomatal closure and lower CO_2_ fixation induced by leaf dehydration and, on the other hand, changes in sink growth depending on sink strength [23]. This sort of change has been recently described in the potato variety under study (Rosanna), despite the fact that this variety has been chosen for its relative drought resistance [20]. Note that, as previously observed [20], the number of tubers was not affected by water stress (data not shown), since stress was induced after onset of tuber swelling. The similar starch and water contents in the control and SWD tubers were consistent with the specific processes of potato tuber growth involving continuous cell proliferation and enlargement.

It is well known that drought has a strong effect on the sucrose-to-hexose ratio in sink organs such as tuber [23]. For the present study, this ratio can be considered a biochemical biomarker of water deficit since stress is known to reduce the degradation of sucrose [24]. Thus, in the growing tuber, there is a cycle of sucrose degradation and resynthesis [25] where the conversion of sucrose to starch represents a possible regulation of the internal osmotic potential.

The myo-inositol content could be considered to be a biochemical biomarker of both water deficit (Figure 3F,G) and plant stress tolerance, as previously reported [26,27]. The study of MIPS sweet potato mutants with low myo-inositol biosynthesis has shown a putative role for this molecule in drought tolerance. Myo-inositol may play a role in osmotic adjustment [28]. Although only trace quantities of myo-inositol were detected in the tissues of the potato tubers, significant differences were observed during growth depending on the water status of the plant (Figure 3G). However, the increase observed in water-stressed tubers may not provide the explanation for this osmotic effect since the content was no more than 1 µM.

So far, it has not been possible to provide an explanation for the concomitant evolution of sucrose content and the acidification of the flesh tissues during tuber filling and under drought conditions. However, it has been reported in a study of orange fruits [29] that such a decrease in the pH of sink tissues may affect carbohydrate partitioning, with acidity affecting both sucrose hydrolysis and vacuolar sequestration.

### 3.2. Relationship between Transverse Relaxation Times and Tissue Features

Results from the present study indicate that the MRI transverse relaxation signal in the three main potato tuber tissue types can distinguish two water pools with relaxation times of around 60–90 ms and above 200 ms. These relaxation signal components can be linked to particular subcellular compartments only in a limited way. Indeed, as mentioned in the Introduction, the diffusive exchange of water between the compartments partly averages the individual relaxation signals and the contributions from different compartments may overlap. Further, the tissues are potentially composed of cell populations with distinct volume distributions that may result in a bi-exponential signal from the vacuole compartment [15,30]. Nevertheless, the results of the current study that provide transverse relaxation parameters measured by MRI and several tuber biochemical descriptors go some way to improve on previously-proposed transverse relaxation signal assignment. An interpretation was proposed by Hills et al. [31] of the relaxation signal obtained using TD-NMR at 100 MHz (Car-Purcell-Meiboom-Gill sequence (CPMG), TE = 0.4 ms) on potato tubers, following investigation of low-temperature tuber samples and analysis of isolated tissue fractions. At 5 °C (corresponding to the temperature of the present study), the transverse relaxation signals consisting of four peaks centered at (i) approximately a few ms, (ii) 10 ms, (iii) 100 ms and (iv) 200–300 ms, and were attributed to the water in the starch granules, cell walls, cytoplasm and vacuoles, respectively. The differences between the TD-NMR protocols used in [31] and the MRI protocols used in the present study offer a partial explanation for the differences in the results. First, the MRI approach did not allow for measurement of the two fast-relaxing T_2_ components recorded by Hills et al. (about a few ms and 10 ms). Second, the differences between the T_2_ values measured in [31] and in the present study were to be expected as several studies have demonstrated that relaxation times are affected by experimental conditions (TE, magnetic field strength, temperature) [32,33]. Nevertheless, on the basis of the T_2_ and relative amplitude values (Figure 5), it can be hypothesized that the two water pools detected by the MRI relaxation signal correspond to either (i) the cytoplasm and vacuole compartments or (ii) the vacuoles of two cell populations with distinct volume distributions. Note that despite the short TE used in the TD-NMR study of fresh tubers reported in [34] (23 MHz (CPMG, TE = 0.3 ms, 25 °C), the fast-relaxing components remained undetected. Indeed, here, only two water pools with a T_2_ of (i) around 60 ms and (ii) around 450–500 ms were distinguishable.

As previously demonstrated for fruit [15,35], the present study shows that MRI relaxometry is effective in revealing differences between tuber tissues, especially those between the pith and the other tissues. Indeed, the T_2_ for component (2) was highest in pith and its relative signal distribution differed from that measured in the cortex and flesh throughout the growth period (Figure 5 and Table 4). Multiple relaxation mechanisms are potentially involved in generating these different multi-exponential transverse relaxation times between the tuber regions, and these probably relate to both the specific composition and the structure of this tissue type. Indeed, the greater cell volume in pith (resulting from the cells’ particular tubular form [5]) compared with that of the cortex and flesh is a likely contributor to the tissue’s higher component (2) T_2_. It has also been reported that pith has a slightly lower starch content than the other potato tissues [7], confirmed in the present study by observation of light micrographs (Appendix A). The impact of starch content on the relaxation times of vacuolar and cytoplasmic water pools was observed in banana fruit during ripening [36] and was explained by the occurrence of chemical exchanges between water and starch molecules in amyloplasts and diffusive water exchanges between the amyloplastic, cytoplasmic and vacuolar compartments. Similarly, the lower starch content observed in the defective perimedullary region of a potato tuber compared with neighboring healthy tissues was related to the higher T_2_ of component (2) [11]. In the present study, T_2_ did not reveal differences between the cortex and flesh, although slight but significant differences in their pH and sucrose contents were observed at 32 DASE. It was not possible to provide an interpretation of the higher relative signal intensity of component (2) in pith compared with the other two tissues.

The relationship between cell size and relaxation time has been demonstrated for stem [14] and fruit tissues [15]. Indeed, in peaches, apples [15] and pear fruit [37], the increase in T_2_ for the component assigned to the vacuolar water pool was related to the increase in cell size in growing fruit. Unexpectedly, in the present study, the developmental stage (expressed in DASE) and, consequently, the increased tuber size only slightly affected component (2) T_2_ for the flesh tissue (Figure 5 and Table 5), whereas the effect of developmental stage was insignificant for component (2) T_2_ in other tissues and for component (1) T_2_. These differences can be explained by the different processes involved in fruit and tuber growth. In fruits, rapid cell expansion is the dominant mechanism of fruit growth. By contrast, potato enlargement is mainly based on cell division occurring in the cambium-like layer of the perimedullary zone [19]. Consequently, cell sizes remained largely unchanged and the perimedullary cells remained smaller than the pith cells. This explains the absence of any major impact of tuber growth on relaxation parameters. It is also consistent with the fact that T_2_ for the control and stressed tubers did not vary significantly at 60 DASE in any of the tissues studied, despite the fact that, at this stage, there was a significant difference in tuber size (Figure 1). The differences between control and SWD conditions observed in component (2) T_2_ measured in the tuber tissues at the intermediate stages (DASE 39 to 53), when the tuber growth rate was greatest, may be due to a delay in the growth kinetics of stressed plants. It also may come from differences in water hydration of tuber tissues (Table 2). Note that water stress may also induce changes in cell membrane permeability that can affect the multi-exponential relaxation signal [13].

The relaxation times for the two water pools detected by MRI were the parameters that clearly reflected the impact of water stress on tuber development (Figure 5B,C) and were the principal parameters explaining the second PCA axis (Figure 6C). The possible impact of cell size and water content on relaxation time under stress conditions requires clarification through further microscopic studies.

## 4. Materials and Methods

### 4.1. Experimental Design and Sampling

The potato tubers analyzed in this study were harvested from potato plants of the Rosanna cultivar of *Solanum tuberosum* potato grown under glass (IGEPP, Le Rheu, France). The variety Rosanna was chosen for its high economical potential due to its relatively low sensitivity to drought. Note that it is still sensitive to drought, according to the fact that all potato plants have a more or less sparse and shallow root system. Pre-germinated tubers of similar weight (average 23 ± 2 g fresh weight) and with 1 to 2 sprouts were selected for planting. Single tubers were placed at 25 cm depth in 25 L plastic pots (Airpot^®^, 27 cm diameter, 50 cm high) filled with a mixture (Falienor^®^ ref. 992016F1) of sandy loam (40% *v/v*) and peat moss (60% *v/v*) with added clay (40 kg m^−3^) and NPK (0.7 kg m^−3^ PG-MIX 14-16-18) (soil solution: pH 5.8 ± 0.2 and Ec (1/1.5) 0.7 ms cm^−1^). During the tuber set, which occurred 14 to 24 days after shoot emergence (DASE), soil humidity levels were maintained at 70% of field capacity for all plants. According to a previous study [20], two water regimes were then applied during the tuber-filling period: (i) a ‘well-watered’ regime, corresponding to soil humidity levels equivalent to 70% of field capacity (control) and (ii) a ‘severe water deficit’ (SWD) regime, corresponding to humidity levels equivalent to 20% of field capacity. The pot weight equivalent to 70% and 20% was calculated using the method described by Earl [38] to define the soil water holding capacity. In the present study, control and SWD conditions corresponded to a relative soil water of about 20% and below 5%, respectively. In SWD conditions, this value was reached 15 days (39 DASE) after water withholding and it was then maintained by daily water input with the water amount to be supplied determined by daily weighing. Under both regimes, watering ceased for top-kill at 63 DASE and final harvesting was carried out 24 days later.

Four plants grown under each water regime (control and SWD) were harvested during the tuberization growth period at 32, 39, 46, 53 and 60 DASE. In the case of SWD plants, these dates corresponded, respectively, to 1 to 5 weeks after irrigation and were withheld. For each plant, only the three heaviest tubers (labelled T01, T02 and T03 in decreasing weight order) were harvested and stored for a few days before further analysis (Figure 1). In total, 24 tubers (2 water regimes; 4 plants per water regime; 3 tubers per plant) were collected at each the previously mentioned DASE. Different tubers from the same plant were analyzed individually. MRI analysis was carried out on whole potato tubers (T02) on all the above harvesting dates (Appendix A). pH and redox potential were measured at 32, 46 and 60 DASE on a 1 cm thick cross section taken from the center of each potato tuber (T01; Appendix A). Biochemical composition (sugar, organic acid and starch contents) was also determined at 32, 46 and 60 DASE. A 5 mm thick cross section was excised from the center of each potato tuber (T03) and the potato tuber tissues were then separated using a scalpel as shown in Appendix A. Potato tuber tissues were frozen in liquid nitrogen and stored at −80 °C, lyophilized, and then ground to obtain a powder. This allowed the water content (WC) to be determined, expressed as a percentage of the tuber fresh weight for each tissue using the following equation:WC (%) = (FW − DW)/FW × 100
where FW and DW represent, respectively, the fresh and dry weights (after lyophylization) of the potato tuber tissues. For some tuber sizes and pith structures, the samples obtained from the pith were not sufficient to determine its biochemical composition. For this reason, physiological and biochemical analysis of the pith was not performed.

### 4.2. MRI Acquisition Protocols and Image Processing

Images of potato tubers were recorded on a 1.5 T MRI scanner (Magnetom Avanto, Siemens, Erlangen, Germany) equipped with a circular polarized head array coil. For the MRI analysis, eight potato tubers were aligned in the center of two rectangular trays (four tubers on each tray; Appendix A). The trays were then inserted in a box placed in the temperature-regulating device installed inside a RF coil designed to conduct MRI acquisitions at 5 ± 1 °C. A 2D Multi-Spin Echo (MSE) sequence [39] was used for the determination of transverse relaxation time (T_2_) in the tissues of the potato tubers. Images were acquired with the following parameters: imaging matrix 160 × 160, field of view 152 mm × 152 mm, slice thickness 5 mm, repetition time 10 s, echo time 6.5 ms, number of echoes 256, bandwidth 290 Hz/pixel and 2 averages.

Multi-exponential T2 parameters were estimated for three different tissue types found in potato tubers, i.e., the cortex, the flesh and the pith. Regions of interest (ROIs) corresponding to each tissue were segmented automatically (Figure 4B,C and Appendix A) using an algorithm developed using Scilab software. At first, tubers were segmented from the background applying the Otsu automatic thresholding method [40]. According to its position in the images and after a first erosion with a 5 × 5 square kernel in order to remove peripheral pixels (skin), a mask is created for each tuber. The cortex was then delimited by a second erosion with a 5 × 5 square kernel (considering the strip of thickness equal to 3 pixels eroded from the contour of the mask). The pith was delineated by considering the central region of the tuber with a threshold applied by a method based on entropy of the histogram [41]. Any segmented objects on the periphery were manually eliminated. The flesh was the region between the cortex and the pith presenting heterogeneous gray levels. The mean signal from each ROI was computed for all MSE sequence images to generate T_2_ decay curves. Values corresponding to a signal-to-noise ratio lower than 7 were removed from the T_2_ curves before fitting, making it possible to assume zero-mean noise distribution in magnitude-reconstructed images at all echo times. Then, T_2_ relaxation signals were fitted according to the Levenberg-Marquardt algorithm using TableCurve software. The optimal number of components to describe the multi-exponential model was selected, taking account of the coefficient of determination (R^2^) of the fit, the global fitting standard error and the errors in all individual parameters. The intensity of individual signal components was expressed as a percentage of the total signal, thereby reflecting the distribution of water between water pools.

### 4.3. Biochemical Composition

Polar components were extracted from potato tuber powders by using methanol, chloroform, and water in succession. The powder (10–15 mg) was mixed for 15 min with 400 µL of methanol containing internal standard (400 µM adonitol). Then, 200 µL of chloroform was added and the mixture was homogenized for 10 min. Water (400 µL) was added and the mixture was stirred vigorously for 20 s. The mixture was then centrifuged for 5 min at 12,000× *g* at 15 °C (model 3–18KC, Sigma). The chloroform phase was discarded and a volume of 50 µL of the methanol-water phase was dried using a SpeedVac system. Dried extracts were stored at −20 °C until analysis. For GC-FID analysis, vacuum-dried aliquots were re-suspended in 50 µL of methoxyaminehydrochloride solution in pyridine (20 mg/mL) and the mixture was stirred for 60 min at 40 °C. Then, 50 µL of N-methyl-N-(trimethylsilyl) trifluoroacetamide was added. Derivatization at 40 °C for 30 min and injection were automatically performed using a MultiPurpose Sampler (MPS, Gerstel). Carbohydrates and organic acids were quantified by GC-FID (6890N, Agilent) with a column Tg-5 ms 30 m × 0.32 mm × 0.25 µm (Thermo Scientific 26098-1430) as described by [42]. Metabolite contents were estimated through reference to the adonitol signal and expressed in µmol/g of potato tissue dry weight.

Starch content was determined using an enzymatic method described by [41]. In brief, potato tuber powder (5–10 mg) was mixed in a 2 mL Eppendorf with 1 mL of a methanol-water solution (1:1, *v/v*) followed by 300 µL chloroform. Two liquid phases were separated from the plant powder after 5 min of centrifugation (13,200 rpm, 4 °C, model 5415R, Eppendorf, Le Pecq, France). The lower phase was recovered and 750 µL of methanol was added and continuously mixed for 20 min at 4 °C. The mixture was then centrifuged at 14,000 rpm at 4 °C for 20 min. The supernatant was decanted off and the starch was dispersed by autoclaving (2 bars, 120 °C, 2 h) and then hydrolyzed with amyloglucosidase (6 IU/tube) in a water bath (56 °C, 1 h 30 min). The glucose content in the supernatant was determined using the microplate method [43]. The starch content was expressed in % of dry weight.

### 4.4. Redox Potential and pH Measurements

Redox potential (Eh) was measured as described by [44] using a Ag/AgCl Reference electrode Radiometer analytical E21M002 and a Radiometer Analytical platinium plate electrode (5 × 5 mm M241 Pt), with a Voltcraft VC850 multimeter (10 × 106 Ohm input resistance). Electrodes were applied 5 mm apart in the cortex and then in the flesh of potato tubers (Appendix A). The value retained corresponded to that displayed on the multimeter when it was stable for one minute. After being measured using the Ag/AgCl reference electrode, all redox potentials were transformed to give Eh according to the “normal hydrogen electrode”. Redox electrodes were calibrated at the start of the measurements and every 10–12 measurements, with Mettler Toledo Redox buffer solution 220 mV (pH = 7) composed of Potassium hexacyanoferrate (III), Potassium hexacyanoferrate (II), Potassium dihydrogen phosphate and Disodium hydrogen phosphate. All measurements were conducted outdoors in an environment identified as being free from electromagnetic interference.

Once the Eh was measured in the different tissues of the potato tubers, portions of the cortex were excised and quickly ground with a pestle and mortar. The mixture was filtered through cotton wool inserted in a 2 mL syringe. The juice was squeezed onto a Horiba LAQUAtwin-pH-22 m for pH measurement. The same protocol was applied to the flesh.

### 4.5. Statistical Analysis

Data obtained from biochemical analysis and from the application of MRI relaxometry to potato tuber tissues were used as input variables in statistical analysis performed with Statgraphics Centurion XVII data mining software (statpoint Technologies, Inc., Warrenton, VA, USA). The Student t-test or one-way analysis of variance (ANOVA) were used to determine whether there were any statistically significant differences between the means of independent groups at a 95% confidence level. The Mann–Whitney test or Kruskal–Wallis test were used as non-parametric alternatives to the Student t-test and ANOVA, respectively, whenever the normal distribution of residuals was not observed. Principal component analysis (PCA) was also used as a non-targeted exploratory analysis to predict the effects of water regime, tissue type and growth stage.

## 5. Conclusions

In the present study, tuber growth was investigated using MRI relaxometry and biochemical parameters, with particular attention paid to the main tissues of the tuber, i.e., the cortex, the flesh and the pith. The study also characterized the differential effect of water regime on the particular composition and growth kinetics of each tissue type. The results indicate that, during the period of tuber enlargement, no major changes specific to tissue type occurred in the biochemical composition of the flesh or cortex. The small number of variations observed were mainly due to the dilution effect of starch accumulation. Nevertheless, variations in the organic compounds made it possible to reveal different patterns in the metabolism of the two tissues that probably relate to the continuous active growing of the flesh. Water deficit affected only the size of the tubers, having no impact on overall tissue functioning, the only exception being the myo-inositol metabolism. On the T2-weighted MRI images, the cortex and the pith appeared clearly as homogeneous regions, whereas the flesh was characterized by a relatively heterogeneous signal. Both T2 values and relative signal intensity allowed distinctions to be made between the pith and the cortex, whereas the flesh could be differentiated from the other two tissue types only through the relative signal intensity. The present study observed no significant changes in T2 at any stage of tuber development. This difference from recorded observations of T2 in fruit is consistent with the specific growth pattern of tubers, consisting in cell proliferation rather than in the increase of final cell size in fruit. In future studies, it could be useful to apply a post-processing scheme that clusters similar voxels according to the multi-exponential relaxation parameters, in order to better discriminate between the tissues [45]. Although MRI relaxation parameters have been shown to act as biomarkers for tuber tissue types and stress response, their interpretation in terms of tissue structure and composition could be further improved via microscopic analysis.

## Figures and Tables

**Figure 1 plants-11-01918-f001:**
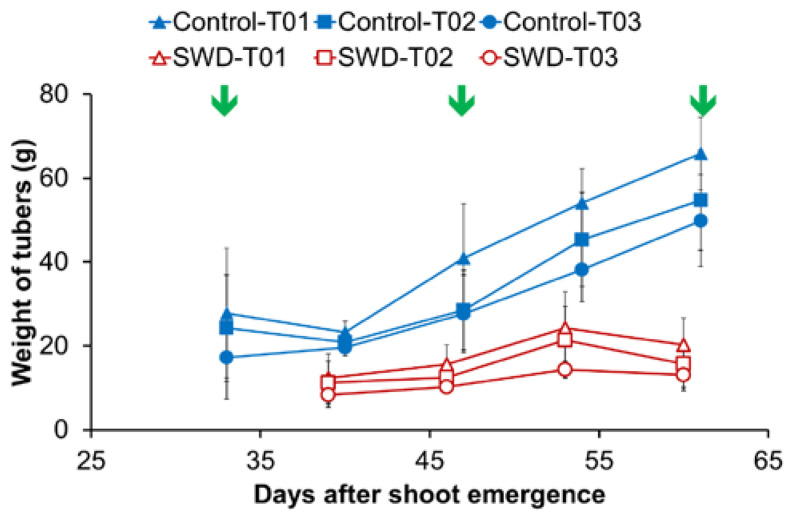
Evolution of tuber weight during the growth period expressed in days after shoot emergence (DASE). For each of the eight plants (4 Control & 4 subjected to severe water deficit SWD), the three heaviest tubers were labeled T01 to T03 in decreasing order of mass at harvest. Weight values correspond to the average of four observations (*n* = 4) and standard deviation are indicated by the segment bars. Arrows indicate the measurement dates for physiological parameters. In the case of SWD plants, dates correspond, respectively, to 1 to 5 weeks after irrigation was withheld.

**Figure 2 plants-11-01918-f002:**
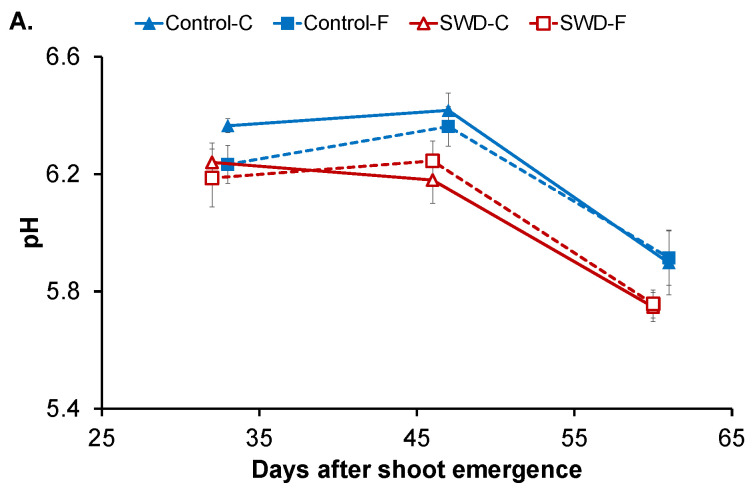
Evolution of (**A**) pH and (**B**) redox potential corrected for pH = 7 (Eh (pH7)) in potato tuber tissues (C: Cortex; F: Flesh) during tuber filling expressed in days after shoot emergence for both control and severe water-deficit (SWD) conditions. Values correspond to the average of four observations (*n* = 4) and standard deviation are indicated by the segment bars. The statistical significance of differences in value for tissues, growth periods and water regimes is reported in Table 1, Table 2 and Table 3, respectively.

**Figure 3 plants-11-01918-f003:**
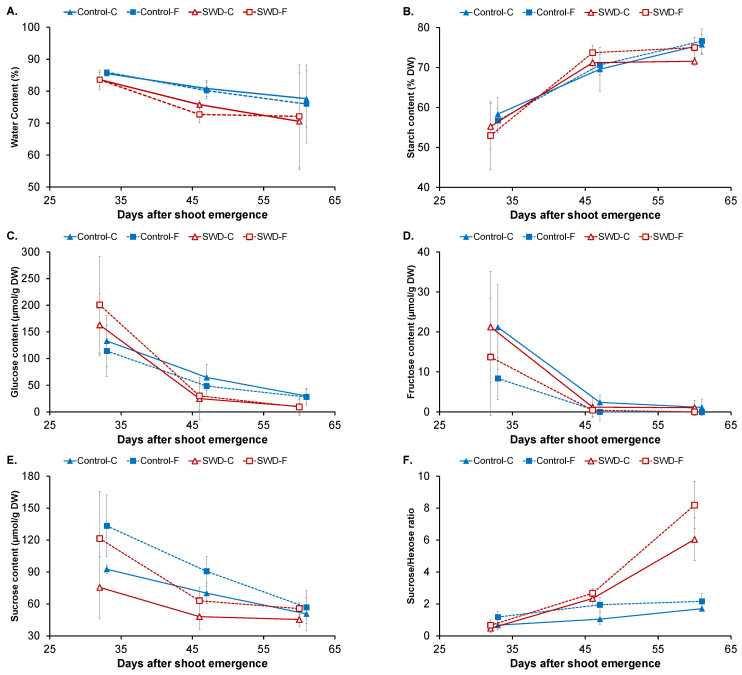
Evolution of (**A**) water content, (**B**) starch content, (**C**) glucose content, (**D**) fructose content, (**E**) sucrose content, (**F**) sucrose-to-hexose ratio, (**G**) myo-inositol content and (**H**) organic acid content in potato tuber tissues (C: Cortex; F: Flesh) during tuber filling expressed in days after shoot emergence for both control and severe water-deficit (SWD) conditions. Values correspond to the average of four observations (*n* = 4) and standard deviation are indicated by the segment bars. The statistical significance of values between tissues, growth periods and water regimes is reported in Table 1, Table 2 and Table 3, respectively.

**Figure 4 plants-11-01918-f004:**
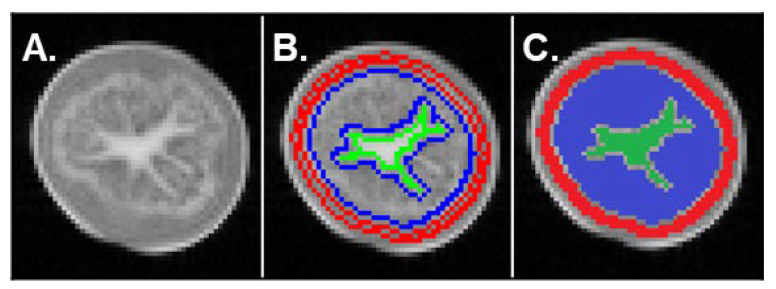
(**A**) MRI image acquired with a Multi-Spin Echo sequence (echo time = 130 ms, repetition time = 10 s, 0.8 × 0.8 × 5 mm^3^ voxel) of a potato tuber at 60 days after shoot emergence (DASE). (**B**) Masks of the three different tissues (cortex, flesh, pith) obtained by means of an automatic segmentation algorithm developed for transverse relaxation time (T_2_) computation. (**C**) The three different regions obtained following segmentation according to T_2_: cortex (red), flesh (blue) and pith (green). See Appendix A for images of the other tubers analyzed at 60 DASE.

**Figure 5 plants-11-01918-f005:**
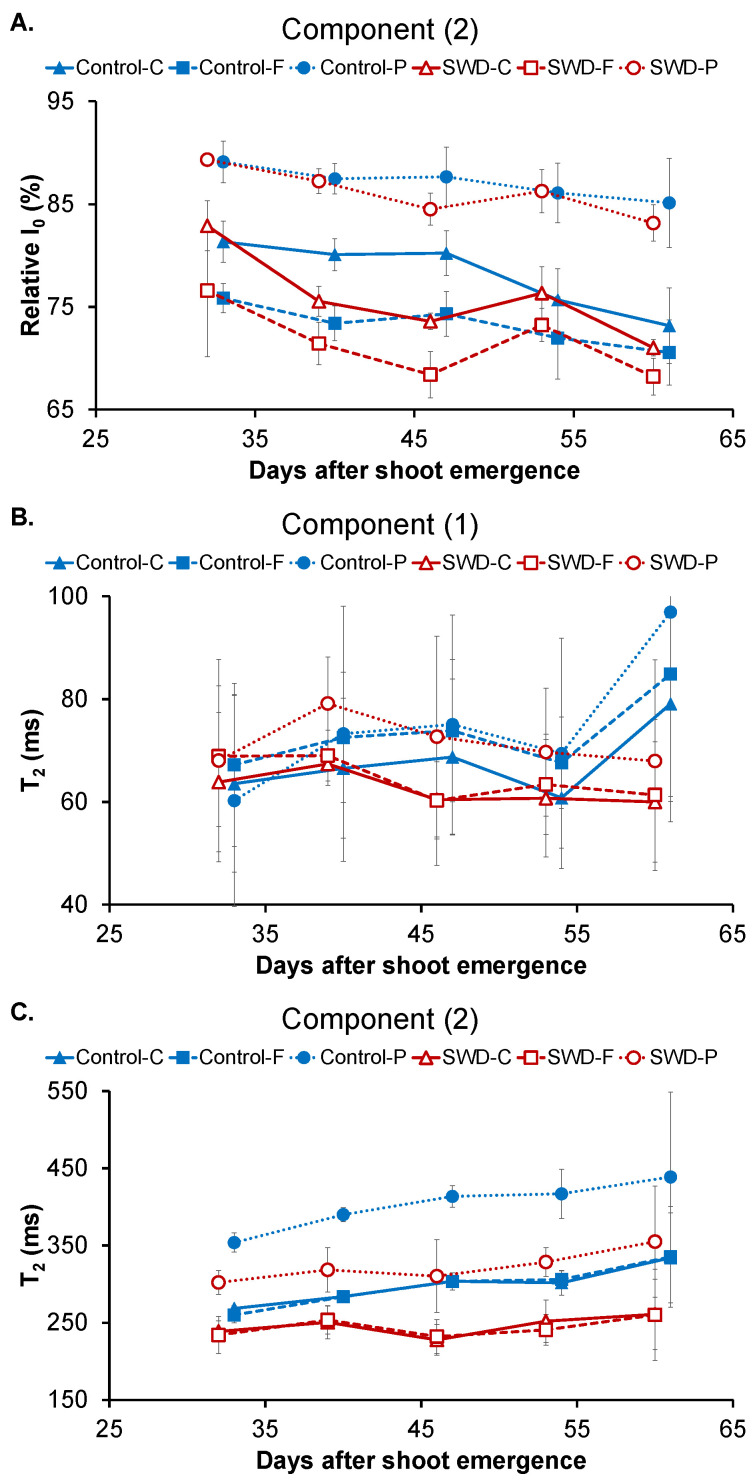
Evolution of (**A**) component (2) relative signal distribution, (**B**) component (1) transverse relaxation time (T_2_) and (**C**) component (2) T_2_ in potato tuber tissues (C: Cortex; F: Flesh; P: Pith) during tuber filling expressed in days after shoot emergence for both control and severe water deficit (SWD) conditions. Values correspond to the average of four observations (*n* = 4) and standard deviation are indicated by the segment bars. The statistical significance of values between tissues, growth periods and water regimes is reported in Table 4, Table 5 and Table 6, respectively.

**Figure 6 plants-11-01918-f006:**
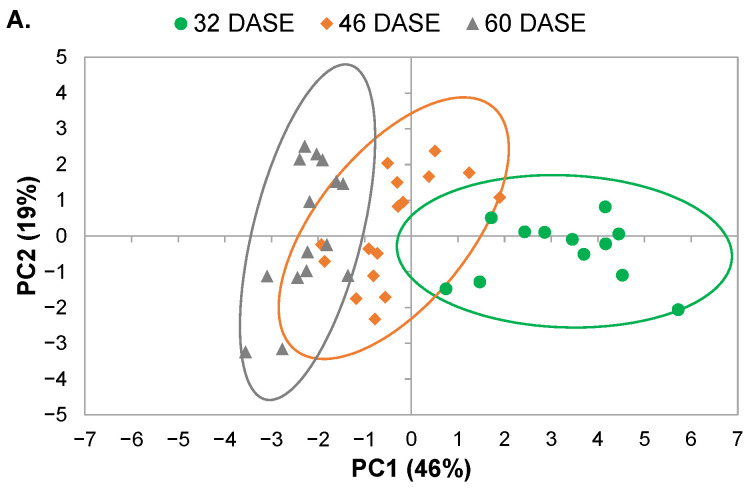
Principal component analysis (PCA) score plots (**A**,**B**) and loading plot for variables (**C**) derived from data obtained from MRI relaxometry and physiological measurements of potato tuber tissues using principal components 1 (PC1) and 2 (PC2). Observations shown on the score plots are differentiated by colour and symbol to indicate sample classification according to (**A**) growth period and (**B**) water regime. Continuous lines on score plots (**A**,**B**) represent the ellipsis of confidence at 95% confidence for each group.

**Table 1 plants-11-01918-t001:** Statistical evaluation of the effect of tissue type on biochemical parameters (Student t-test or Mann–Whitney test; Statgraphics, alpha = 5%). Different letters between Cortex and Flesh for the same DASE indicate a statistically significant difference.

DASE	32		46		60	
Control						
Variables	Cortex	Flesh	Cortex	Flesh	Cortex	Flesh
pH	**a**	**b**	a	a	a	a
Eh (pH7)	a	a	a	a	a	a
WC	a	a	a	a	a	a
Starch	a	a	a	a	a	a
Glucose	a	a	a	a	a	a
Fructose	a	a	a	a	a	a
Sucrose	**a**	**b**	a	a	a	a
S/(G + F)	a	a	**a**	**b**	a	a
Myo-inositol	-	-	-	-	a	a
Organic acids	**a**	**b**	a	a	**a**	**b**
SWD						
Variables	Cortex	Flesh	Cortex	Flesh	Cortex	Flesh
pH	a	a	a	a	a	a
Eh (pH7)	a	a	a	a	a	a
WC	a	a	a	a	a	a
Starch	a	a	a	a	**a**	**b**
Glucose	a	a	a	a	a	a
Fructose	a	a	a	a	a	a
Sucrose	a	a	**a**	**b**	a	a
S/(G + F)	a	a	a	a	a	a
Myo-inositol	a	a	a	a	a	a
Organic acids	**a**	**b**	**a**	**b**	**a**	**b**

**Table 2 plants-11-01918-t002:** Statistical evaluation of the effect of water regime on biochemical parameters (Student t-test or Mann–Whitney test; Statgraphics, alpha = 5%). Different letters between Control and SWD conditions for the same DASE indicate a statistically significant difference.

DASE	32		46		60	
Cortex						
Variables	Control	SWD	Control	SWD	Control	SWD
pH	**a**	**b**	**a**	**b**	a	a
Eh (pH7)	a	a	a	a	a	a
WC	a	a	**a**	**b**	a	a
Starch	a	a	a	a	a	a
Glucose	a	a	**a**	**b**	**a**	**b**
Fructose	a	a	a	a	a	a
Sucrose	a	a	a	a	a	a
S/(G + F)	a	a	**a**	**b**	**a**	**b**
Myo-inositol	-	-	-	-	a	a
Organic acids	**a**	**b**	**a**	**b**	**a**	**b**
Flesh						
Variables	Control	SWD	Control	SWD	Control	SWD
pH	a	a	**a**	**b**	**a**	**b**
Eh (pH7)	a	a	a	a	a	a
WC	a	a	**a**	**b**	a	a
Starch	a	a	a	a	a	a
Glucose	a	a	a	a	**a**	**b**
Fructose	a	a	-	-	-	-
Sucrose	a	a	a	a	a	a
S/(G + F)	a	a	a	a	**a**	**b**
Myo-inositol	**a**	**b**	-	-	a	a
Organic acids	a	a	a	a	a	a

**Table 3 plants-11-01918-t003:** Statistical evaluation of the effect of growth phase on biochemical parameters (ANOVA test followed by the Tukey HSD test, or the Kruskal–Wallis test followed by the Bonferroni test; Statgraphics, alpha = 5%). Different letters between 32, 46 and 60 DASE for the same growth condition indicate a statistically significant difference.

DASE	Control			SWD		
Cortex						
Variables	32	46	60	32	46	60
pH	**a**	**a**	**b**	**a**	**a**	**b**
Eh (pH7)	**a**	**ab**	**b**	**a**	**b**	**c**
WC	a	a	a	a	a	a
Starch	**a**	**ab**	**b**	**a**	**b**	**b**
Glucose	**a**	**b**	**b**	**a**	**ab**	**b**
Fructose	**a**	**ab**	**b**	a	a	a
Sucrose	a	a	a	a	a	a
S/(G + F)	**a**	**a**	**b**	**a**	**a**	**b**
Myo-inositol	-	-	-	a	a	a
Organic acids	**a**	**b**	**b**	**a**	**b**	**b**
Flesh						
Variables	32	46	60	32	46	60
pH	**a**	**a**	**b**	**a**	**a**	**b**
Eh (pH7)	**a**	**b**	**c**	**a**	**a**	**b**
WC	a	a	a	a	a	a
Starch	**a**	**b**	**b**	**a**	**b**	**b**
Glucose	**a**	**b**	**b**	**a**	**ab**	**b**
Fructose	-	-	-	-	-	-
Sucrose	**a**	**b**	**b**	**a**	**ab**	**b**
S/(G + F)	**a**	**b**	**b**	**a**	**a**	**b**
Myo-inositol	**a**	-	**b**	a	a	a
Organic acids	**a**	**b**	**b**	**a**	**b**	**b**

**Table 4 plants-11-01918-t004:** Statistical evaluation of the effect of tissue type on MRI parameters (ANOVA test followed by the Tukey HSD test or the Kruskal–Wallis test followed by the Bonferroni test; Statgraphics, alpha = 5%). Different letters between Pith, Cortex and Flesh for the same DASE indicate a statistically significant difference.

DASE	32			39			46			53			60		
Control															
Variables	Pith (P)	Cortex (C)	Flesh (F)	P	C	F	P	C	F	P	C	F	P	C	F
I_0_ (comp 2)	**a**	**b**	**c**	**a**	**b**	**c**	**a**	**b**	**c**	**a**	**b**	**b**	**a**	**b**	**b**
T_2_ (comp 1)	a	a	a	a	a	a	a	a	a	a	a	a	a	a	a
T_2_ (comp 2)	**a**	**b**	**b**	**a**	**b**	**b**	**a**	**b**	**b**	**a**	**b**	**b**	a	a	a
SWD															
Variables	Pith (P)	Cortex (C)	Flesh (F)	P	C	F	P	C	F	P	C	F	P	C	F
I_0_ (comp 2)	a	a	a	**a**	**ab**	**b**	**a**	**b**	**c**	**a**	**b**	**b**	**a**	**ab**	**b**
T_2_ (comp 1)	a	a	a	a	a	a	a	a	a	a	a	a	a	a	a
T_2_ (comp 2)	**a**	**b**	**b**	**a**	**b**	**b**	**a**	**b**	**b**	**a**	**b**	**b**	a	a	a

**Table 5 plants-11-01918-t005:** Statistical evaluation of the effect of water regime on MRI parameters (Student t-test or Mann–Whitney test; Statgraphics, alpha = 5%). Different letters between Control and SWD conditions for the same DASE indicate a statistically significant difference.

DASE	32		39		46		53		60	
Pith										
Variables	Control	SWD	Control	SWD	Control	SWD	Control	SWD	Control	SWD
I_0_ (comp 2)	a	a	a	a	a	a	a	a	a	a
T_2_ (comp 1)	a	a	a	a	a	a	a	a	a	a
T_2_ (comp 2)	**a**	**b**	**a**	**b**	**a**	**b**	**a**	**b**	a	a
Cortex										
Variables	Control	SWD	Control	SWD	Control	SWD	Control	SWD	Control	SWD
I_0_ (comp 2)	a	a	a	a	a	a	a	a	a	a
T_2_ (comp 1)	a	a	**a**	**b**	**a**	**b**	a	a	a	a
T_2_ (comp 2)	a	a	**a**	**b**	**a**	**b**	**a**	**b**	a	a
Flesh										
Variables	Control	SWD	Control	SWD	Control	SWD	Control	SWD	Control	SWD
I_0_ (comp 2)	a	a	a	a	a	a	a	a	a	a
T_2_ (comp 1)	a	a	a	a	a	a	a	a	a	a
T_2_ (comp 2)	a	a	**a**	**b**	**a**	**b**	**a**	**b**	a	a

**Table 6 plants-11-01918-t006:** Statistical evaluation of the effect of growth phase on MRI parameters (ANOVA test followed by the Tukey HSD test or the Kruskal–Wallis test followed by the Bonferroni test; Statgraphics, alpha = 5%). Different letters between 32, 39, 46, 53 and 60 DASE for the same growth condition indicate a statistically significant difference.

DASE	Control					SWD				
Pith										
Variables	32	39	46	53	60	32	39	46	53	60
I_0_ (comp 2)	a	a	a	a	a	**a**	**ab**	**bc**	**abc**	**c**
T_2_ (comp 1)	a	a	a	a	a	a	a	a	a	a
T_2_ (comp 2)	a	a	a	a	a	a	a	a	a	a
Cortex										
Variables	32	39	46	53	60	32	39	46	53	60
I_0_ (comp 2)	**a**	**ab**	**ab**	**bc**	**c**	**a**	**ab**	**ab**	**ab**	**b**
T_2_ (comp 1)	a	a	a	a	a	a	a	a	a	a
T_2_ (comp 2)	a	a	a	a	a	a	a	a	a	a
Flesh										
Variables	32	39	46	53	60	32	39	46	53	60
I_0_ (comp 2)	a	a	a	a	a	a	a	a	a	a
T_2_ (comp 1)	a	a	a	a	a	a	a	a	a	a
T_2_ (comp 2)	**a**	**ab**	**ab**	**ab**	**b**	a	a	a	a	a

## Data Availability

Not applicable.

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
