# Peer review of "Characterization of the Water Shortage Effects on Potato Tuber Tissues during Growth Using MRI Relaxometry and Biochemical Parameters"

_plants, 2022, doi:10.3390/plants11151918_

Round 1

Reviewer 1 Report

The manuscript describes the response of potato tubers on water stress. Main instrument is the statistical analysis of the relations between different biochemical markers and MRI relaxation times. The main conclusion is that the water deficit stress does not change the tissue types, but reduces the tuber size. Concerning the MRI images, the reader would profit from a Figure showing the relaxation time maps in addition to the T2 weighted image in Fig. 6 and the supplement, since these are the basis for the further analysis. Also selected time curves for sample regions of pith, flesh and cortex together with the bi-exponential fits would help with the assessment of the data. Maybe in the supplement.

Summarized, the paper is worth to be published with some minor corrections:

l 63: "gravity": you mean mass?

l65: the term kinetics may be replaced by "rate".

l69: Specify "water dynamics". MRI can determine water dynamics on different scales. Relaxation is controlled by diffusion processes on microsecond to millisecond scales, time series on seconds and longer.

l 72f: What is the relation between water content and T1, T2, D?

l92: relaxation times in plant tissues are mainly controlled by cell sizes. What is the relation to starch content?

Figure 1: The number of samples is not clear. 3 tubers for each of 4 plants: n = 12?

Tables 1 and 2: Describe the entries a, b, A, and the meaning of the colors (red, black)

l238ff:

The effect of water stress on the long T2 component is obvious in contrast to the short component. Possible interpretation of the relaxation time are discussed later. However, tentative interpretations (long = vacuole, short = cytoplasm?) should be stated already here to facilitate the understanding.

Figure 6: Is the echo time really tE = 130 ms? This would mean a heavy T2-weighting by suppressing the short component (T2 ca. 70 ms). Is this the reason of the clear discrimination of the different tissues?

Further, effective T2 may decrease effectively with longer tE due to diffusion in possible internal gradients. The fact that there are images for tE = 130 ms might indicate that there are no internal gradients, i.e. no air bubbles in the tubers. Later, l511, the authors state that T2 is not different in cortex and flesh (for tE = 7.4ms!). Apparently the long echo time makes a differentiation possible.

This all should be explicitly discussed in the text. Perhaps with the help of selected echo train curves.

l287f: Please state clearly the number of samples: 3x4 per regime per harvesting date?

l314: Why 5°? What were the environmental conditions during the growth period?

l321: Could you describe the automatic segmentation in more detail? What were the criteria for segmentation?

l500f: is this supported by the photographs in the supplement, Figure S2?

l516 an l519: is this a contradiction? [34] stated a correlation between long T2 and dry matter. In line 519 the authors state lower dry matter with long T2 component. Please clarify.

l542&568: It is good to point out the added value by microscopic studies.

Reviewer 2 Report

The article  refers to characterization of the water shortage effects on potato tuber tissues during growth under water deficit using MRI relaxometry and chosen biochemical parameters. Authors propose MRI parameters as biomarkers for the assessment of tuber development.

Some improvement of paper are required. In M&M I did not find information how the soil drought was maintained – when it started, how it was measured, how many days cessation of watering was needed to reach 20% FWC, how did the authors checked field water capacity. If it was not monitored, you could not claim that it was severe drought.

Is it something known about tolerance of cv. Rosanna for drought, is it susceptible or resistant. Authors wrote in Discussion that that the number of tubers was not affected by water stress. Please show that data together with the FW of tubers.

The conducted research is very interesting and using MRI for evaluation of potato tuber development sounded promising, but after reading whole article I am not convinced if it is true. Authors showed that starch content correlated negatively with component I0 and that the most significant variables according to PCA were T2 values obtained from MRI measurements (Figure 5C). What does it mean, please explain – could we predict some biochemical parameters of tubes based on T2 value? Authors also wrote that MRI relaxation parameters have been shown to act as biomarkers for tuber tissue types and stress response. For tuber tissue types ok, but for stress response, please convince readers.

Other comments:

Supplementary Fig. S4 it does not bring anything new comparing with Fig. 6, so in my opinion is not necessary.

Figures show mean values with what standard deviation or standard error?, please add this info in the description of figures as well as n=… .

Table 1 is not fully described – the description is only for part A. Also there is no explanation what means small characters  (a, b etc.) and what capital letters (A).

Which means were compared in Table 1 A, B and Table 2 B, if only cortex and flesh or control and SWD, so you should use rather t-Student test to compare two means rather than Tukey test.

Having so many variables did whether it would be better to do multivariate analysis of variance instead of one-way analysis of variance.

Round 2

Reviewer 2 Report

The authors took into account the reviewer's suggestions and corrected the manuscript. I have no further comments, I recommend the manuscript for publication.